# Lytic and Latent Genetic Diversity of the Epstein–Barr Virus Reveals Raji-Related Variants from Southeastern Brazil Associated with Recombination Markers

**DOI:** 10.3390/ijms25095002

**Published:** 2024-05-03

**Authors:** Paula D. Alves, Paulo Rohan, Rocio Hassan, Eliana Abdelhay

**Affiliations:** 1Oncovirology Laboratory, Division of Specialized Laboratories, Instituto Nacional de Câncer (INCA), Rio de Janeiro 20230-130, RJ, Brazil; 2Stem Cell Laboratory, Division of Specialized Laboratories, Instituto Nacional de Câncer (INCA), Rio de Janeiro 20230-130, RJ, Brazil

**Keywords:** Epstein–Barr virus, oncovirus, genetic diversity, latent, lytic, BZLF1, LMP1

## Abstract

Epstein–Barr virus (EBV) is a ubiquitous gammaherpesvirus etiologically associated with benign and malignant diseases. Since the pathogenic mechanisms of EBV are not fully understood, understanding EBV genetic diversity is an ongoing goal. Therefore, the present work describes the genetic diversity of the lytic gene *BZLF1* in a sampling of 70 EBV-positive cases from southeastern Brazil. Additionally, together with the genetic regions previously characterized, the aim of the present study was to determine the impact of viral genetic factors that may influence EBV genetic diversity. Accordingly, the phylogenetic analysis of the *BZLF1* indicated two main clades with high support, BZ-A and BZ-B (PP > 0.85). Thus, the BZ-A clade was the most diverse clade associated with the main polymorphisms investigated, including the haplotype Type 1 + V3 (*p* < 0.001). Furthermore, the multigene phylogenetic analysis (MLA) between *BZLF1* and the oncogene *LMP1* showed specific clusters, revealing haplotypic segregation that previous single-gene phylogenies from both genes failed to demonstrate. Surprisingly, the *LMP1* Raji-related variant clusters were shown to be more diverse, associated with BZ-A/B and the Type 2/1 + V3 haplotypes. Finally, due to the high haplotypic diversity of the Raji-related variants, the number of DNA recombination-inducing motifs (DRIMs) was evaluated within the different clusters defined by the MLA. Similarly, the haplotype BZ-A + Raji was shown to harbor a greater number of DRIMs (*p* < 0.001). These results call attention to the high haplotype diversity of EBV in southeast Brazil and strengthen the hypothesis of the recombinant potential of South American Raji-related variants via the *LMP1* oncogene.

## 1. Introduction

The Epstein–Barr virus (EBV) is a ubiquitous gammaherpesvirus that is well adapted to humans; however, it is causally associated with benign and malignant lymphoproliferations [1]. Although the tumorigenic process is not part of its replicative cycle, the constant reactivation of EBV breaks this equilibrium stage with the host and can be associated with an increased risk of developing EBV-associated diseases [1,2]. This long-term coevolution with the host leads to the complex persistence dynamics of EBV [1]. EBV is commonly transmitted via saliva, and after passing through the oropharyngeal epithelium, it infects circulating B cells, which can establish a latent infection and persist in memory B cells during the host lifespan [3]. EBV replication can occur through the proliferation of infected B cells or virion production during the lytic cycle [4]. In primary infection, the lytic cycle is incomplete since the main immediate viral lytic transactivator, BZLF-1 or the Zta protein, needs the entire EBV genome to be methylated to initiate the productive cycle. Consequently, EBV can begin its complete lytic cycle only after latency is established and viral reactivation occurs [4].

The EBV virion has a linear double-stranded DNA genome of approximately 172 kb and has the potential to translate more than 80 proteins and 46 functional small untranslated RNAs [5]. During latency, only eleven RNAs are transcribed, of which only nine are translated. However, in the lytic cycle, the majority of EBV transcripts are expressed [5,6]. Furthermore, the EBV genome has several internal repetitive regions with short and long sequences [1]. Due to its genomic complexity, historically, the classification of EBV genetic diversity was established mainly through subgenomic regions, focusing mainly on latent genes that exhibit differential expression in various EBV-associated tumors and may be related to the development of neoplasms [6,7]. Importantly, the factors that influence EBV genetic diversity are point mutations, deletions and insertions, and homologous recombination [8,9,10]. The major genetic variation in EBV is based on differences in the *EBNA* genes, which classify EBV into Types 1 and 2 [6]. However, only recently, in a haplotypic association with the V3 variant of the promoter zone (Zp) of the lytic transactivator BZLF-1, Type 1 was directly associated with an increased risk of developing disease to confer a functional increase in viral lytic reactivation [11]. Additionally, previous epidemiological evidence revealed the presence of the Type 1 + V3 haplotype in tumors from Southeast Asia and AIDS-associated lymphoma [11,12]. Notably, the linkage of the EBV types and Zp is already expected due to the proximity of the *BZLF1* and *EBNA3* loci [8,9].

Unlike its promoter, the genetic diversity of the *BZLF1* gene is not well understood [13,14]. *BZLF1* is the master regulator of the lytic cycle, and its expression in latent cells is sufficient to initiate and complete the lytic cycle [4]. Moreover, several properties, such as immunomodulation capacity, proliferative signaling, and death resistance mechanisms, have been directly associated with this lytic transactivator, indicating its central role in the biology of EBV. Despite this, its genetic diversity has not been well described, and the methods used to study it are considerably different [13,14].

Regarding South American EBV variants, a recent study from our group [15] described two main *LMP1* oncogene variants circulating in southeastern Brazil through phylogenetic analysis. The Med-pattern variants clustered in the clade mainly consisted of asymptomatic individuals and were associated with the haplotype Type 1 + V1. Conversely, Raji-related variants presented more lymphomas in their clade than Med-pattern variants did, and Raji-related variants were demonstrated to be highly diverse by harboring diverse haplotypes, including Type 1 + V1/V3 and Type 2 + V3. Similarly, South American variants related to the Raji strain were previously suggested to be recombinant variants [16]. However, this previous *LMP1* phylogeny did not show a clear distribution of these haplotypes (Type 1 + V1/V3 or Type 2 + V3) within clades, which makes it difficult to understand their relationship with the studied variants. Therefore, characterizing the phylogenetic diversity of *LMP1* alone was insufficient for understanding the viral genetic linkage between the EBV variants circulating in the Brazilian population [15].

Comparing the latent genes to lytic genes, such as *LMP1* and *BZLF1*, respectively, the latter have lower nucleotide variation and, consequently, a reduced phylogenetic signal [6,8,13,16]. Nonetheless, the low variation may illustrate ancient strain relationships that could be obscured when evaluating more variable regions of the genome today. Therefore, the use of multigene evolutionary reconstruction containing the coding region of both genes may allow us to understand haplotypic relationships that single-gene approaches would not. Bearing these considerations in mind, the aim of the present work was to describe *BZLF1* genetic diversity in Brazilian samples, to perform a multigene phylogenetic analysis of *BZLF1* and *LMP1*, to characterize lytic and latent viral haplotypes from the same samples, and, ultimately, to investigate whether highly diverse EBV variants are associated with recombination markers.

## 2. Results

### 2.1. Characteristics of the EBV-Positive Samples

The EBV-positive samples included a total of 70 individuals, including a group of cases as follows: 13 classic Hodgkin lymphoma (cHL) patients, with a median age of 27 years and a sex ratio of M:F = 3.33:1; 6 Burkitt lymphoma (BL) patients, with a median age of 4.5 years and a sex ratio of M:F = 2:1; 6 posttransplant EBV reactivation (PTER) patients, with a median age of 30.5 years and a M:F sex ratio = 5:1; and 45 asymptomatic carrier (AC) donors, with a median age of 26 years and a sex ratio of M:F = 1:2. From the AC, four donors did not have Brazil as their country of origin; thus, their home country was considered representative of EBV origin.

### 2.2. Genetic Diversity of the BZLF1 Lytic Gene Circulating in Southeastern Brazil

The phylogenetic tree reconstructed through Bayesian inference indicated two main clades (Figure 1), BZ-A and BZ-B, with high support (PP = 0.85 and PP = 1, respectively). The BZ-A clade was demonstrated to be most diverse due to the greater number of nucleotide changes. Among the obtained *BZLF1* sequences, 27% were in the major BZ-A clade and 73% were in the BZ-B clade. Within the BZ-A clade, three minor clades were identified: BZ-A1 (PP = 0.54), BZ-A2 (PP = 1), and BZ-A3 (PP = 0.78). BZ-A1 comprised 17% of the new BZ-A sequences from the BZ-A clade and was characterized by geographic references from Oceania (including the lineage Wewak2), Africa, and South America (BR). The BZ-A2 clade was established only by Asian reference sequences (including the GD1, GD2, HKNPC1, Akata, and M81 lineages) and did not group any of the new obtained *BZLF1* sequences. Finally, the BZ-A3 clade was established mainly by African geographic reference sequences, including the AG876 and Jijoye lineages. Moreover, BZ-A3 encompassed most of the new Brazilian sequences from the BZ-A clade (*n* = 83%).

The BZ-B clade was principally divided into three clades with high support (PP > 0.82): BZ-B1, BZ-B2, and BZ-B3 (Figure 1). The BZ-B1 clade included mostly new obtained BZLF1 sequences from the BZ-B clade (86.5%) and was characterized by geographic references from America, Africa, and Europe, including the lineages B95-8 (EBV1 prototype), Raji, Daudi, and Mutu. Within BZ-B1, two additional clades, BZ-B2 (PP = 1) and BZ-B3 (PP = 0.88), clustered more sequences with greater support. The BZ-B2 clade was composed of geographical references from Asia and Oceania and did not contain any of the new obtained *BZLF1* sequences. The BZ-B3 clade was established by African and European references and grouped 9.6% of the new *BZLF1* sequences within the BZ-B clade. Additionally, some *BZLF1* sequences from Brazil, including two new sequences (3.9%), were gathered with sequences from East Asia and Oceania at the base of the major BZ-B clade.

In the *BZLF1*-based phylogenetic tree, it was possible to observe the haplotype relationship with the type and Zp variant established in 65 sequences. Consistent with the nucleotide changes observed in the phylogeny, the BZ-A clade was more diverse, with all the haplotypes for Type 2 and V3 (T2V3) and most of the haplotypes for Type 1 and V3 (T1V3) from the BZLF1-based phylogenetic tree. Within clade BZ-A, T2V3 and T1V3 represented 78% and 22%, respectively, of the sequences, with total absence of the Type 1 and V1 (T1V1) haplotype. In contrast, in the BZ-B clade, 98% of the sequences were T1V1 and 2% of the sequences were T1V3, with a total absence of the T2V3 haplotype (Figure 1; Appendix A). In summary, through the evolutionary history of the *BZLF1* gene, the present work demonstrated that the oldest BZ-A clade is the most diverse clade, harboring the most nucleotide changes and a haplotypic diversity from sequences that are not present in the phylogeny. However, BZ-B was demonstrated to be less diverse and related to the T1V1 haplotype.

### 2.3. Polymorphic Diversity of the New Obtained BZLF1 Sequences and Its Relationship with Variants Classified through Phylogenetic Analysis

The polymorphic diversity of the Zta protein revealed a total of 11 polymorphisms present in the new *BZLF1* sequences obtained in the present work (Figure 2). The new *BZLF1* sequences in the BZ-A clade presented greater genetic diversity, sheltering 10 of the 11 described polymorphisms (*p* < 0.05). Of these, eight represented 100% of the sequences in the BZ-A clade (T68A, S76P, T124P, V146A, V152A, Q163L, Q195H, and A205S). The other polymorphisms were heterogeneously distributed between the sequences, with E176D being present in 84% of the sequences and with the deletion of an aa at position 127 (del127) present in 10.5% of the new sequences in the BZ-A clade. Conversely, of the BZ-B sequences, only 4 of the 11 polymorphisms had a heterogeneous distribution among the new *BZLF1* sequences: these included del127 (47%), A206S (8%), V146A (4%), and A205S (2%). Notably, the A206S polymorphism was exclusive to the BZ-B clade.

Next, the polymorphisms were identified in the different functional domains of the Zta protein. The protein transactivation domain (TAD) comprises amino acids (aa) 1 to 167 and was the domain that presented the highest number of polymorphisms, containing 64% of all the polymorphisms found. The dimerization domain (DD), which comprises aa 196 to 221, harbored 18% of the polymorphisms. Both the regulatory (RD) and the DNA-binding (DBD) domains, comprising aa 168 to 177 and 178 to 195, respectively, harbored 9% of the polymorphisms together. The relationship of the domains with the established clades revealed a set of polymorphisms (T68A, S76P, T124P, V152A, and Q163L) in the TADs exclusive to the BZ-A clade (*p* < 0.0001). Similarly, the V146A polymorphism from the TAD was present mainly in the BZ-A population (*p* < 0.0001). In contrast, del127 was more common in the BZ-B subgroup (*p* = 0.005). In the DD cohort, the A205 polymorphism was present mainly in the BZ-A population (*p* < 0.0001). However, A206S was detected only in BZ-B, the only polymorphism exclusive to this clade. The only polymorphism observed in RD, E176D, was found exclusively in the BZ-A clade (*p* < 0.0001). Finally, the DBD sheltered the Q195H polymorphism exclusively in BZLF-1 variants from the BZ-A clade (*p* < 0.0001). Notably, all sequences that presented the V146A and A206S polymorphisms in BZ-B belonged to the BZ-B3 subclade, indicating a possible marker polymorphism for this subclade (*p* = 0.016 and *p* > 0.001, respectively), which appears to have the most recent origin. Thus, in general, the observation of polymorphisms allows us to understand the phylogenetic division of the main clades of the *BZLF1* gene, revealing greater genetic diversity in the BZ-A clade than in the BZ-B clade (Figure 2A,B). Finally, there was no correlation between the polymorphisms or the *BZLF1* clades and clinical outcomes (*p* > 0.05).

### 2.4. Multigene Analysis of BZLF1 and LMP1 Reveals the Associations of Other Lytic and Latent Targets

The topology of the multigene phylogenetic tree mainly resembled the topology of the phylogenetic tree based on the *LMP1* oncogene alone; this observation was expected because the genetic variation in this gene was greater than that in the *BZLF1* lytic gene. However, the haplotypic-specific segregation of sequences that were not present in the analysis was surprisingly not expected. In this way, multigene analysis was able to demonstrate the linkage of lytic and latent targets from sequences present in the analysis but also from viral genetic regions that were absent (Figure 3).

The multigene tree of *LMP1* and *BZLF1* generated using the Bayesian inference method presented two major clades with equally high support (PP = 0.94). Within both major clades, the formation of clades with high support (PP > 0.9) comprising the classic EBV lineages was clearly observed. Major clade I (MCI) contained fewer sequences, including 8% of the newly analyzed sequences. In MCI, 75% of the sequences belonged to BZ-A clade and 25% belonged to BZ-B clade. The sequences related to BZ-A clade included sequences from East Asia and Oceania, while the BZ-B clade included sequences from all continents. Furthermore, the sequences classified as BZ-A + LMP1-Unclassified variants (LMP1-UC) from Brazilian sampling clustered paraphyletic to Asian strains, together with the only Brazilian sequence linked directly to the Asian sequences at the *LMP1* level (ACBR30_BZ-B_BR_China2-pattern_T1V3).

Major clade II (MCII) contained most of the sequences analyzed, which included 92% of the Brazilian sequences from the present work. From MCII, two subclades arose separating a few Asian sequences (minor subclade, PP = 1) from major subclades related to diverse sequences from all continents, but mainly from Africa and the Americas (PP = 0.81). Notably, a dyad of Brazilian samples was grouped with high support (PP = 0.97) at the base of this major subclade (taxon in gray and italics in Figure 3). The dyad remained without a clear classification in the multigene analysis, which was also the case for the *LMP1* phylogeny for these samples, despite belonging to the BZ-B clade in the *BZLF1* phylogeny (Appendix A). Then, paraphyletic to the dyad, a clade with high support (PP = 0.94) arose segregating a large group of samples in a subclade with high support (PP = 0.93) from a minor clade related to the EBV prototype T1V1 (PP = 1). This minor clade related to the EBV prototype clustered a few sequences from South America (BR), Europe (UK), and Africa with the same haplotype relationship (BZ-B + LMP1 B95-8 + T1V1). The main subclade paraphyletic to the EBV prototype subclade is the one from which all the main genetic diversity of the Brazilian sequences emerges.

This substantial subclade paraphyletic to the EBV T1 prototype included 85% of all the newly analyzed samples and geographic references from all continents, including the lineages AG876 (EBV T2 prototype), Jijoye, Mutu, Daudi, and Raji. First, the segregation of the two main LMP1 variants circulating in southeastern Brazil is clear (the Mediterranean pattern (Med-pattern) and the Raji strains). Second, as expected from multigene analysis, within the distribution of LMP1 variants, the homogeneous distribution of variants related to the *BZLF1* sequence (BZ-A and BZ-B) in clades of high support was notable (PP > 0.9).

Despite the heterogeneous diversity of this substantial subclade, the multigene phylogenetic tree of *BZLF1* and *LMP1* revealed specific clusters which a previous single-gene phylogeny failed to demonstrate. Surprisingly, these clusters are related to the segregation of other lytic and latent target haplotypes, for which the following sequences were absent in this analysis: T1V1, T1V3, and T2V3. Notably, compared with T1V1, the haplotype associated with an increased risk of viral lytic reactivation and therefore associated with pathogenic potential, T1V3, was shown to be associated with sequences that had more nucleotide changes in the tree from both genes analyzed according to multigene analysis. Importantly, only Brazilian samples with the BZ-A + China2-pattern and BZ-A + Raji-variant haplotype were associated with T1V3. The clusters of the BZ-B + B95-8/Med pattern were associated with V3 of the ZP variant only with EBV type 2. These results suggest more genetic diversity in the Brazilian samples related to the Raji variant than in those related to the B95-8/Med pattern.

### 2.5. DNA Recombination-Inducing Motifs Are Associated with Highly Polymorphic EBV Haplotypes and Clinical Outcomes

In light of the convergent results, which indicate a greater diversity of some viral haplotypes in relation to others, and considering that phylogenetic reconstruction alone is not able to fully explain EBV genetic diversity, we turned our attention to the presence of DNA recombination-inducing motifs (DRIMs), which could explain, at least in part, these highly diverse EBV variants (Figure 3 and Figure 4). For this purpose, the main haplotype variants of EBV circulating in southeastern Brazil were chosen (BZ-A + Med pattern, BZ-B + Med pattern, BZ-A + Raji, and BZ-B + Raji, *n* = 43). However, among the DRIMs evaluated, only the canonical meiotic recombination motif (CCTCCCCT) was absent in any of the sequences. The chi-like recombination sequence (TGGTGG) was significantly different between the different haplotypes (*p* = 0.004). Additionally, when comparing the haplotypes pair by pair, the BZ-A + Med pattern haplotype had a significantly lower amount of DRIMs than the BZ-A + Raji haplotype (*p* = 0.01), BZ-B + Med pattern (*p* = 0.003), and BZ-B + Raji (*p* = 0.006) (Figure 4).

The motifs recognized as recombination initiators, TGGAG and CCCAG, exhibited significant differences between the haplotypes (*p* < 0.001 and *p* < 0.001, respectively). The TGGAG motif was present in greater amounts within the BZ-A + Raji haplotype than in the BZ-B + Med (*p* < 0.001) and BZ-B + Raji (*p* < 0.001) haplotypes. The second haplotype with the highest amount of this DRIM was BZ-A + Med, followed by BZ-B + Med (*p* = 0.037) and BZ-B + Raji (*p* = 0.045), as shown in Figure 4B. This finding demonstrates that sequences from the BZ-A clade directly influence the amount of this DRIM in the haplotype. The CCCAG motif was significantly different between the haplotype groups (*p* < 0.001), mainly because of the greater number of this motif in the BZ-A + Raji haplotype compared to the BZ-B + Med pattern (*p* = 0.001). This difference was also observed between the BZ-B + Raji and BZ-B + Med haplotypes (*p* = 0.002); unlike in previous observations, this difference seems to be directly influenced by the Raji variant sequences (Figure 4C).

The motif of the human immunoglobulin class switch, GGGCT, was significantly different between the haplotypes (*p* = 0.001) and was less common in the BZ-A + Raji than in the BZ-A + Med (*p* = 0.067), BZ-B + Raji (*p* = 0.014), and BZ-B + Med haplotypes (*p* < 0.001). GGGCT was the only DRIM in which this phenomenon occurred (Figure 4E). The motif related to stop synthesis via DNA polymerase α, AGGAG, also had a differential occurrence between haplotypes (*p* = 0.001), with the BZ-B + Med haplotype having the lowest amount versus BZ-A + Raji (*p* = 0.003) and BZ-B + Raji (*p* = 0.002), as shown in Figure 4D. Moreover, when examining the total amount of DRIMs, we observed that the BZ-A + Raji haplotype had the highest amount of DRIMs, while the BZ-B + Med haplotype had the lowest amount. A difference between these haplotypes was observed (*p* < 0.001), but the difference between individual haplotypes was significant only when the haplotypes BZ-A + Raji and BZ-B + Med were compared (*p* = 0.001), as well as when BZ-B + Raji and BZ-A + Med were compared (*p* = 0.030), as shown in Figure 4F. To conclude, the only DRIM associated with a clinical outcome was the CCCAG motif, which was significantly increased in EBV-positive patients (*p* = 0.022) (Figure 5A). In consonance, when observing the total amount of recombination motifs, the number of recombination motifs was significantly greater in EBV patients than in AC patients (*p* = 0.011) (Figure 5B).

## 3. Discussion

The EBV strains circulating in southeastern Brazil, specifically obtained from sampling in the state of Rio de Janeiro, exhibited high genetic diversity in the *BZLF1* lytic gene, grouping the sequences into two main clades, BZ-A and BZ-B. The BZ-A clade was the most diverse in terms of nucleotide changes, polymorphisms, and viral haplotypes. In addition, according to the multigene phylogeny based on *BZLF1* and *LMP1*, the BZ-A sequences also demonstrated increased *LMP1* diversity. Subsequently, unlike previous single-gene phylogenies (*BZLF1*—this present work and *LMP1* in Alves et al. [15]), specific clusters were observed in the multigene phylogeny, which revealed genetic linkages between lytic and latent targets whose sequences were present (*BZLF1* and *LMP1*) or absent (type classification by *EBNA3C* and Zp variant). Moreover, phylogenetic and SNP analyses may not be sufficient to explain the genetic linkage of different genetic targets that are distant in the EBV genome [8,9,10]. Thus, the number of DNA recombination-inducing motifs (DRIMs) that were previously suggested to contribute to the genetic diversity of EBVs [17,18] was also evaluated. Strikingly, in an unprecedented way, the *BZLF1* and *LMP1* variants most associated with polymorphic and haplotypic diversity were also associated with a greater number of total DRIMs and associated separately with specific motifs that precede a greater number of recombination events in the EBV genome [17]. Moreover, although no clinical outcome has been directly associated with a specific clade or haplotype, interestingly, EBV-positive cases were associated with a greater number of DRIMs than AC donors.

This work is the first comprehensive description of the genetic diversity and classification of the *BZLF1* gene in Brazil, which is one of the most variable lytic genes in the EBV genome. Therefore, the *BZLF1* phylogeny divided our sequences into a more diverse clade (BZ-A) and a less diverse clade (BZ-B). Noteworthily, not all sequences that were clustered in BZ-A (related to Asian strains) by the *BZLF1* gene were grouped in the Asian clade by *LMP1* from the same sample in previous analysis [15]. Similarly, the polymorphisms observed in the sequences of the BZ-A clade have been described as associated with Asian reference sequences [13,19]. Furthermore, in the present work, no clinical outcome was associated with the *BZLF1* clade. In fact, few studies have explored the genetic diversity of the *BZLF1* gene separately or associated its genetic diversity with clinical outcomes [13,14,19]. However, previous work [13] characterized the genetic diversity of the Zta protein in Argentinean sequences and revealed a group of eight polymorphisms (T68A, S76P, T124P, V146A, V152A, Q163L, Q195H, and A205S), all of which were found to be associated with EBV+ pediatric cHL when compared to a group with infectious mononucleosis. The lack of sample stratification with a larger representative number of specific pathologies may be a bias from our present work, which could obscure certain correlations. However, in agreement with the results of the present study, a previous study investigating Chinese samples reported a similar phylogeny, and the same polymorphisms were described here; moreover, no associations with neoplasm cases were found [19].

Because EBV is so complex, its genetic diversity cannot be fully understood by analyzing only a single gene [6,20,21,22,23]. Therefore, the evaluation of *BZLF1* was extended to its promoter zone (Zp) and the viral type. Consequently, an exclusive haplotypic association was found between the BZ-A clade and the V3 variant of Zp. Hence, BZ-A was related mainly to EBV Type 2 and, to a minor extent, to the haplotype T1V3 in Brazilian samples. In contrast, BZ-B was composed entirely of the haplotype T1V1, which represents virtually all the sequences except for one sequence from North America, which was T1V3. This result was certainly expected because of the proximity of the *BZLF1* to the *EBNA3* locus [8,9]. On the other hand, it was not expected that Brazilian samples previously unrelated to Asian strains by the *LMP1* oncogene would group into the “Asian” BZ-A clade [15].

Taken together with the previous results indicating greater genetic diversity and linkages with lytic and latent targets, multigenetic phylogenetic reconstruction of the same sample based on *BZLF1* and *LMP1* was performed to better understand the relationships between lytic and latent targets. The *LMP1* sequences used were previously classified by our previous work [15] and showed that variants related to the Raji strain and Med-pattern may equally share approximately 80% of the total variants circulating in southeastern Brazil [15]. Importantly, in the same work, variants related to the Raji strain clustered more lymphoma cases in its clade than Med-pattern variants did; that group included mainly AC samples. Furthermore, South American variants related to the Raji strain were significantly associated with the haplotype T1V3. In this way, we focused on these two aforementioned *LMP1* variants due to their frequency, relationship with Brazilian malignant samples, and correlation with the haplotype of pathogenic potential. Accordingly, the major clade I (MCI) and major clade II (MCI) from the multigenetic phylogenetic tree of *BZLF1* and *LMP1* were also observed in both single-gene phylogenies from *BZLF1* (this work) and *LMP1* [15]. Afterwards, the main distinct haplotypes in the multigene tree were defined. Interestingly, the *LMP1* Med-pattern variants exhibited a preferential distribution according to the BZ-B clade classification, while the Raji variant exhibited a significant distribution according to the BZ-A and BZ-B clade classifications. This result was unexpected since the *LMP1* Raji and Med-pattern variants have a nearly homogeneous distribution within these samples [15]. However, Raji-related variants were related not only to both the *BZLF1* clades, but to different haplotypes of the EBV (Raji + T1V1 or T2V3 or T1V3), unlike Med pattern variants related to BZ-B and T1V1. These results together reenforce the hypothesis that South American Raji-related variants may have greater recombinant capacity than Med-pattern variants [15]. Therefore, as previously suggested [8], the different distributions of T1V1 and T1V3 among different EBV strains may reflect the extensive historical recombination of EBV [9]. Conversely, the original African Raji strain is not associated with high haplotypic diversity when evaluating the Zp and *BZLF1* genetic regions, and its haplotype is evaluated as BZ-B and T1V1 [15,16,24].

The present study is the first to describe the high haplotype diversity among lytic and latent targets associated with a greater number of samples of Raji-related variants. In previous studies, Raji variants circulating in South America were associated with unique synapomorphies and were suggested to be new possible recombinants with African variants and Asian-related variants due to their polymorphic markers within *LMP1* oncogene [15,16,20]. However, in the present multigene analysis, it was possible to observe an African sample harboring a BZ-A + Raji + T2V3 haplotype that was also different from the original Raji strain. This result may be related to one of the limitations of our work: the choice of rooting the multigene phylogenetic tree at the midpoint as an estimate of the directionality of the evolutionary process, due to the lack of an ideal *LMP1* sequence to be used as a root. Accordingly, the absence of an ideal outgroup for *LMP1*, as well as the lack of sequences of African origin related to the Raji strain that could demonstrate (or not) BZ-A + Raji + T1V3 in Africa, makes it difficult to refute or confirm the null hypothesis that these South American variants related to the Raji strain are recombinants that emerged in the Americas as a consequence of modern human migration [15,16].

In summary, our results suggest that the Raji-related variants, recently described as the main *LMP1* variants circulating in South America [15,16,20], are a group of complex variants that can be further divided into different viral haplotypic profiles, which seem to have distinct characteristics. Although these profiles were not directly associated with clinical outcomes in the present study, the increase in sample size, as well as the stratification of different types of EBV-associated diseases, warrant new studies that could improve the understanding of EBV genetic diversity.

Moreover, it is known that the specific segregation of different haplotypes of lytic and latent targets that are distant from the EBV genome cannot be explained only by phylogenetic analyses related to point mutational events [8,9,10]; therefore, it was necessary to evaluate genetic markers that may favor recombination events between the different variants and haplotypes [17,18]. We assessed whether recombination-associated markers could be related to EBV variants with high haplotype diversity. Consequently, six DRIMs related to gammaherpesviruses were identified [18]. Consistent with previous results, the BZ-A + Raji haplotype was demonstrated to have the highest total amount of DRIM, especially related to the TGGAG and CCCAG motifs. Despite the apparently heterogeneous distribution of certain DRIMs among different haplotypes, both recombinant primer sequences that were increased in the BZ-A + Raji haplotype have been mutually described to precede the greatest changes in recombination signals and the ratio of occurrence of recombination events in the EBV genome, indicating a direct relationship between the two [17]. Thus, these unprecedented results concerning recombination motifs associated with a specific EBV variant suggest that recombination may have a greater impact on EBV recombination processes. This finding agrees with the described origin of Raji variants in South America and may explain, at least in part, the presence of the T1 + V3 haplotype in these variants.

Furthermore, interesting results were obtained when relating DRIM to different clinical outcomes. Among the six DRIMs evaluated, a recombination initiator (CCCAG) was significantly increased in cases when compared to AC. Additionally, the total DRIM was significantly different between the two groups, with EBV+ patients presenting the highest DRIM compared to AC samples. However, the possible relationship between DRIM and oncogenic processes is still unclear. A possible mechanism was observed for EBV in the P3HR-1 lineage, in which intramolecular recombination events lead to abnormal lytic replication and interruption of viral latency, which results in a high level of antibody titer [25]. In this vein, it is known that mechanisms related to unbalanced lytic reactivation could occur not only in posttransplant patients, perhaps due to immune imbalance, but also with the appearance of lymphomas [26,27]. Additionally, the recombination process is widely described as being associated with DNA replication in other herpesviruses, and mechanisms related to differences in viral replication are expected to emerge [28].

To conclude, these results should be interpreted with caution since this was a preliminary study related to these recombination markers, and it is necessary to further characterize the functionality and not just the presence of these DRIMs in these strains. An important limitation of our study is that we did not use complete EBV genomes, limiting the possibility that recombination tests can be performed using other viral targets. However, the complete genome of EBV has still not been fully sequenced in different populations from diverse geographic regions [29,30]. Therefore, the multigene approach remains an important tool for understanding the genetic diversity of complex viruses, such as EBV, which can provide useful information about genetic diversity and divergence that might otherwise be obscured [21,22].

## 4. Materials and Methods

### 4.1. EBV-Positive Sample Selection

EBV-positive biopsies of 70 individuals from Rio de Janeiro (southeast Brazil) were included in the study as follows: 19 fresh lymph node tissue samples from lymphoma patients (classic Hodgkin lymphoma or cHL, *n* = 13 and Burkitt lymphoma or BL, *n* = 6), 6 peripheral blood mononuclear cell (PBMC) samples from posttransplant EBV reactivation (PTER) patients; and 45 asymptomatic EBV+ carrier (AC) saliva samples, collected as described previously [15]. Patients with lymphomas and PTER were diagnosed at the Integrated Division of Pathology (DIPAT) of the INCA during the period 1995–2019 according to morphological and immunohistochemical criteria established by the WHO Classification [31]. Clinical and demographic data of all individuals, such as age and sex, were obtained from records at the INCA. This study was approved by the Ethics and Research Committee of the INCA (CAAE 3.56999916.5.0000.5274 and 53571116.4.0000.5274).

### 4.2. DNA Extraction

DNA extraction from fresh tissue and saliva samples was performed as described previously [15]. PBMCs were isolated using Ficoll–Histopaque reagent (Sigma-Aldrich, MO, USA) for density gradient separation. The PBMCs were counted in a Neubauer chamber, and the proportion was adjusted to 10–20 × 10^6^ cells for each 1 mL aliquot of DNAzol reagent. Subsequently, the DNA was extracted according to the manufacturer’s protocol and stored at 2 to 8 °C.

### 4.3. EBV Target Identification

The presence of Type 1 or 2 EBV was assessed via polymerase chain reaction (PCR) of the *EBNA3C* gene, as previously described [32]. From the lytic targets, the promoter zone (Zp) and the coding region (*BZLF1*) from the lytic transactivator Zta were amplified using PCR as previously described [13,33]. Moreover, the products from the PCRs of the Zp and BZLF1 regions were purified using QIAquick PCR purification (QIAGEN, Hilden, Germany) following the manufacturer’s protocol. Then, the products were sequenced using Big Dye Terminator v.3.1 (Applied Biosystems, Waltham, MA, USA) in an automated Genetic Analyzer ABI 3130xl (Applied Biosystems, Waltham, MA, USA) at the INCA.

To explore the linkage relationship between the lytic and latent targets of EBV, 53 of the 70 samples with the *BZLF1* sequence that were previously characterized by the *LMP1* oncogene by our group [15] were used together with the *BZLF1* sequences for further multigene analysis (Appendix A).

### 4.4. Zp and BZLF1 Sequence Analysis and Polymorphism Characterization

The Zp and *BZLF1* gene sequences were aligned with their respective reference sequences using the B95-8 prototype (GenBank number NC_007605) as a reference. Visualization, quality assessment, and editing were performed using the SeqMan Pro (DNASTAR, version 11.1.0; Madison, WI, USA). For the classification of the Zp promoter, the sequences were aligned with the reference sequences through the ClustalW algorithm using the MEGA 11 program [34]. After alignment, the sequences were classified following preestablished nucleotide substitution criteria [33] using the AliView v1.28 program.

The Zp-V1 variant is the sequence present in the B95-8 prototype. The V3 variant differs in three positions from the prototype sequence [−141 (A>G), −106 (A>G), and −100 (T>G)], and the positions of these changes are related to the *BZLF1* transcription start site [13]. The analysis and recognition of polymorphisms in *BZLF1* were compared with those of the B95-8 prototype using the AliView v1.28 program, and the results were described according to the functional domain of the protein [13].

### 4.5. BZLF1 Phylogenetic Analysis

For the phylogenetic analysis based on the coding region of the *BZLF1* gene, a database was built with sequences and reference sequences representing different geographic regions obtained from GenBank (Appendix A). This database was aligned using the ClustalW algorithm in the MEGA 11 program [34] for identification and removal of introns.

The reconstruction of the evolutionary relationship was performed using the Bayesian inference method [35] in the MrBayes program. The evolution model used was GTR + I + G, based on the Akaike information criterion (AIC) and selected by the MrModeltest V2 program [36]. The parameters used for the phylogenetic reconstruction were 100 million steps through the Monte Carlo method via Markov chains (MCMC). Additionally, the convergence diagnostic parameters were considered adequate for assessing the subsequent samples obtained through the MCMC method when the estimated sample size (ESS) was greater than 200 [35]. The support of the branches was established by the posterior probability determined using the Bayesian inference method [37]. The generated trees were rooted using an outgroup with the coding region sequence of the Lymphocryptovirus macaca *BZLF1* gene/pfe-lcl-E3 (GenBank number NC_055142). Subsequently, the generated tree was evaluated, visualized, and annotated using the FigTree program (http://tree.bio.ed.ac.uk/software/figtree/, accessed on 12 September 2022).

### 4.6. Multigene Phylogenetic Reconstruction of the Lytic BZLF1 and Latent LMP1 Genes

Initially, two datasets were constructed containing separate alignments for each target, *LMP1* and *BZLF1*. A total of 53 sequences of the *LMP1* coding region previously obtained by our group [15] were subsequently aligned with the *LMP1* coding sequences of the same geographic references previously used for *BZLF1* phylogeny (Appendix A). However, from the 70 references, only 66 had the entire *LMP1* sequence available. The resulting alignment of 119 sequences was assembled using the ClustalW algorithm in MEGA 11. The model selected for the *LMP1* dataset was GTR + G, based on the AIC calculated using MrModeltest V2 [36]. The alignment of the coding region of the *BZLF1* gene obtained for the previous phylogenetic reconstruction was subsequently altered to include the same paired sequences as the *LMP1* sequences (Appendix A). Subsequently, the sequences of the two alignments were concatenated using the SequenceMatrix v1.8.1 program (http://www.ggvaidya.com/taxondna/, accessed on 19 September 2022), resulting in a single multigene alignment, where nucleotides from position 1–738 represented the gene *BZLF1* and nucleotides 739–1803 represented the *LMP1* gene. The resulting alignment file was inputted to the MrBayes program, where two partitions were created—one for each gene and their respective positions—and the specific evolutionary models GTR + I + G for the *BZLF1* and GTR +G for the *LMP1* were selected. The multigene phylogenetic reconstruction was then performed using the Bayesian inference method, with the parameters set to 100 million steps through the Monte Carlo method via Markov chains (MCMC: Markov Chain Monte Carlo), with four different estimates, seeking an average standard deviation less than < 0.01, indicating convergence between the estimates. Afterwards, the resulting parameters were analyzed through convergence diagnosis and were considered adequate when the estimated sample size (ESS) was greater than 200 [37]. The support of the branches was established through posterior probability (PP), which refers to the probability that a given branch is correct, assuming that the model is correct.

### 4.7. Detection of DNA Recombination-Inducing Motifs

To identify the presence of DNA recombination-inducing motifs (DRIMs) within the different haplotypes between BZLF1 and LMP1, DRIMs were identified across the entire length of both coding sequences as well as within their reverse length through a search, as previously described for EBV [17,18]. The location and quantification of these motifs were determined using the Find Motifs (FM) tool in the MEGA11 program [34]. The DRIMs analyzed are described in Appendix A.

### 4.8. Statistical Analysis

The presence of associations between categorical variables was assessed using Fisher’s exact test, and the Bonferroni correction was used when necessary. When comparing two groups of independent samples, the Mann–Whitney U test was used. For comparisons of more than two groups, the Kruskal–Wallis test was chosen. Thus, a *p* value lower than 0.05 was considered to indicate statistical significance. All the statistical analyses were performed using SPSS version 26 (IBM) and R version 4.2.1.

## Figures and Tables

**Figure 1 ijms-25-05002-f001:**
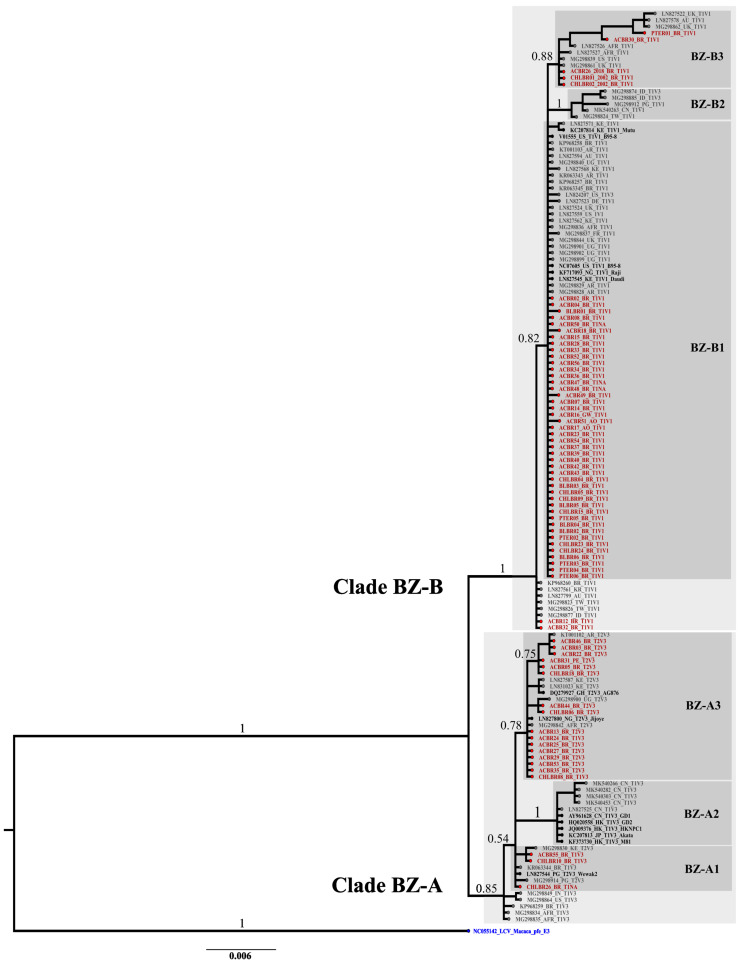
Phylogenetic tree of the new obtained *BZLF1* coding sequences obtained through Bayesian inference. The taxon of all the sequences represents ID_country-origin_Type_Zp_lineage (the last, when applicable). The taxon of the geographic reference sequences included in the analysis (*n* = 71) are in gray, except for the lineage origin marked in black, which includes the Type 1 prototype (GenBank number NC007605) and Type 2 prototype (GenBank number DQ279927) reference sequences. The sequences depicted in red are the newly obtained *BZLF1* sequences (*n* = 70). Major clades BZ-A and BZ-B are marked in light gray, and the main minor clades are highlighted in dark gray. Posterior probability (PP) support values for major branches are indicated, and branch lengths indicate nucleotide substitutions per site according to the scale indicated at the bottom. The roots of the tree are highlighted in blue and were established by the outgroup represented by the coding sequence of the *BZLF1* gene/pfe-lcl-E3 (GenBank number NC_055142) from Lymphocryptovirus Macaca virus. BZLF1: BamHI Z left frame; AFR: Africa; AR: Argentina; AU: Australia; BR: Brazil; CN: China; DE: Germany; FR: France; GH: Ghana; HK: Hong Kong; ID: Indonesia; IN: India; JP: Japan; KE: Kenya; KR: South Korea; NG: Nigeria; PG: Papua New Guinea; TW: Taiwan; UG: Uganda; UK: United Kingdom; US: United States.

**Figure 2 ijms-25-05002-f002:**
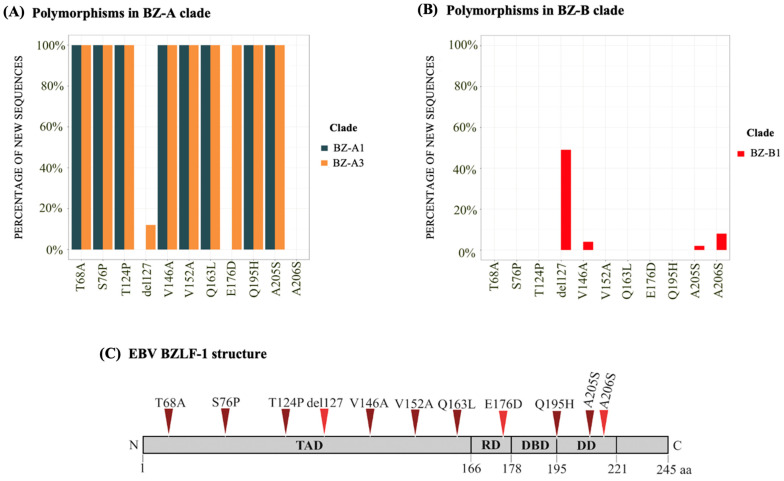
Distribution of the polymorphisms within the BZ-A and BZ-B clades and in the BZLF-1 structure. The *y*-axis of the graph represents, as a percentage, the amount of samples within the clades that had a given polymorphism, while the *x*-axis represents each of the observed polymorphisms. (**A**) Presence of polymorphisms within the BZ-A clade according to the subclades. The blue color represents the BZ-A1 clade, and the orange color represents the BZ-A3 clade. (**B**) Presence of polymorphisms within the BZ-B clade according to the subclades. The red color represents the BZ-B1 clade. The BZ-2 and BZ-3 clades are absent due to the absence of these polymorphisms. (**C**) Structure of BZLF-1 showing the protein domains. The triangles indicate the approximate position of each analyzed polymorphism, colored according to the associated clade (legend on the position of each analyzed polymorphism, colored according to the presence in Brazilian samples, where dark red represents polymorphisms that are present in all sequences from BZ-A, and the red polymorphisms may not be present in sequences from both clades). BZLF1: BamHI Z left frame; del: deletion; A: alanine; E: glutamic acid; F: phenylalanine; G: glycine; K: lysine; L: leucine; M: methionine; P: proline; Q: glutamine; A: arginine; S: serine; T: threonine; V: valine. TAD: transactivation domain; RD: regulatory domain; DBD: DNA-binding domain; DD: dimerization dimming domain; aa: amino acid; N: amino terminus; C: carboxy-terminal domain.

**Figure 3 ijms-25-05002-f003:**
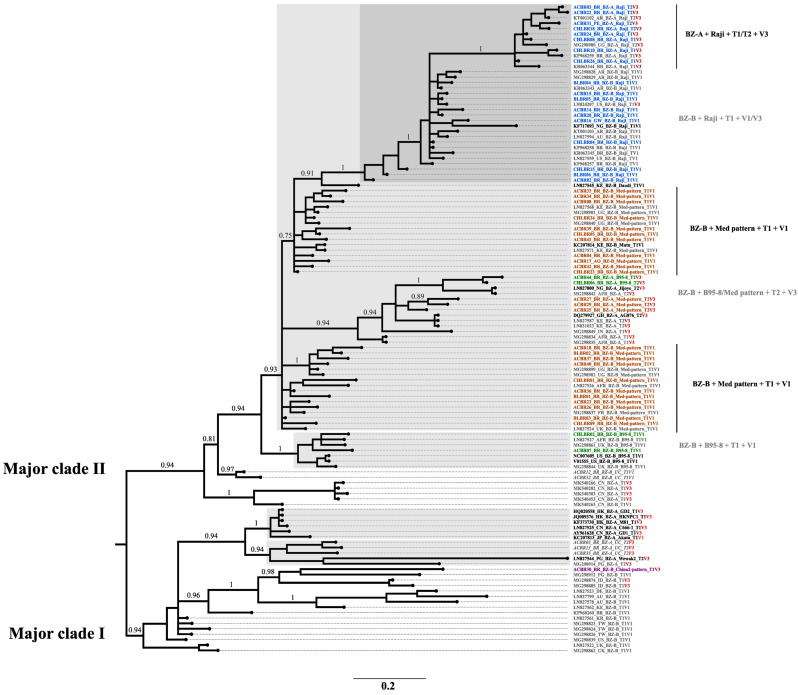
Multigenic phylogenetic tree of the coding sequences *BZLF1* and *LMP1* obtained through Bayesian inference. The taxon of all the sequences represents ID_country_BZLF1_LMP1_lineage-name_Type_Zp (the *LMP1* variant and lineage name were present when applicable). The taxa of the geographic reference sequences included in the analysis (*n* = 65) are in gray, except for the lineage origin marked in black. The taxon of the sequences obtained by our group from southeast Brazil (*n* = 53) is depicted according to its haplotypic relationship with the *BZLF1* and *LMP1* genes as follows: pink (BZ-B + China2 pattern), gray and italic (BZ-A/B + Unclassified or UC), green (BZ-B/A + B95-8), orange (BZ-B/A + Med pattern), and blue (BZ-B/A + Raji). The V3-Zp variants present in the multigenic tree are highlighted in red. All clades related to EBV strains are highlighted in gray or dark gray. Posterior probability (PP) support values for major branches are indicated, and branch lengths indicate nucleotide substitutions per site according to the scale indicated at the bottom. On the right side of the tree, the main haplotypes of the new Brazilian sequences are highlighted in black or gray. BZLF1: BamHI Z left frame; LMP1: latent membrane protein 1; Zp: promoter zone of BZLF1; Med: Mediterranean; PP: posterior probability; AFR: Africa; AR: Argentina; AU: Australia; BR: Brazil; CN: China; DE: Germany; FR: France; GH: Ghana; HK: Hong Kong; ID: Indonesia; IN: India; JP: Japan; KE: Kenya; KR: South Korea; NG: Nigeria; PG: Papua New Guinea; TW: Taiwan; UG: Uganda; UK: United Kingdom; US: United States.

**Figure 4 ijms-25-05002-f004:**
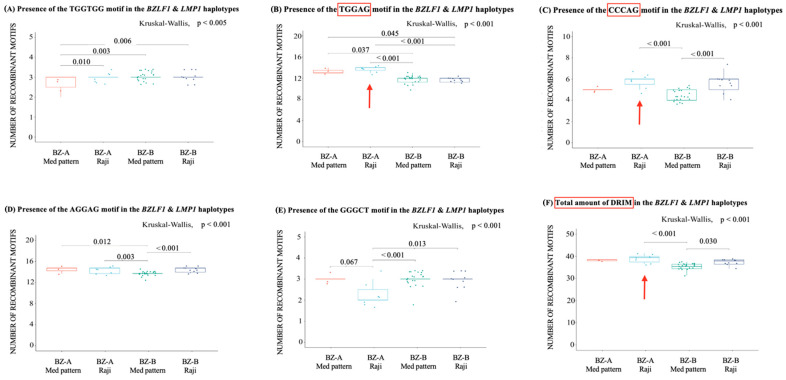
Quantitative evaluation of the DNA recombination-inducing motifs in the main EBV *BZLF1* and *LMP1* haplotypes. The *y*-axis of each graph demonstrates the number of recombinatory motifs, while the *x*-axis represents the 4 haplotypes of BZLF1 + LMP1. Kruskal–Wallis tests were performed for all comparisons, and when significant, the Mann–Whitney test was used for pairwise comparisons with Bonferroni correction. Only those genes whose *p* value was significant according to this test are indicated by the bars in the graphs. The darker line represents the median, while the dots represent the count of each sequence. With the addition of random variation, the dots do not overlap. The red color in the boxes represents the BZ-A + Med pattern haplotype, the light blue color represents the BZ-A + Raji haplotype, the green color represents the BZ-B + Med pattern haplotype, and the dark blue color represents the BZ-B + Raji haplotype. (**A**) Comparison between the haplotypes regarding the presence of the chi-like recombination sequence (TGGTGG); (**B**) comparison between the haplotypes regarding the presence of the sequence found as a recombination primer (TGGAG); (**C**) comparison between the haplotypes regarding the presence of the sequence found as a recombination primer (CCCAG); (**D**) comparison between haplotypes regarding the presence of the human immunoglobulin class switch sequence (GGGCT); (**E**) comparison between the haplotypes regarding the presence of the sequence known to interrupt synthesis via DNA polymerase α (AGGAG); (**F**) comparison between haplotypes regarding the total presence of recombination motifs. DRIM: DNA recombination-inducing motif; BZLF1: BamHI Z left frame; LMP1: latent membrane protein 1; T: thymine; G: guanine; A: adenine; C: cytosine.

**Figure 5 ijms-25-05002-f005:**
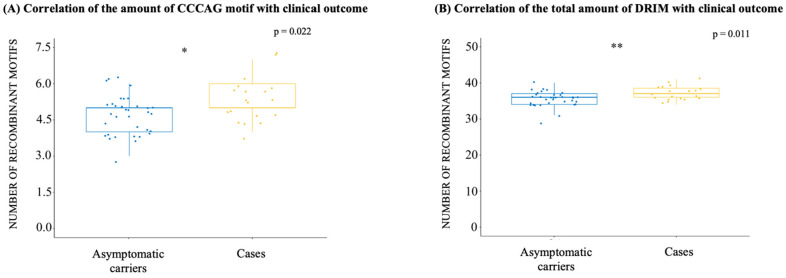
DNA recombination-inducing motifs associated with clinical outcomes. The *y*-axis of each graph demonstrates the number of recombinant motifs, while the *x*-axis represents the two groups studied in the present work: asymptomatic carriers and EBV patients. The darker line represents the median, while the dots represent the count of each sequence, with random variation added so that the points do not overlap. The blue color represents the AC group, and the yellow color represents the case group. The Mann–Whitney U test was performed for comparisons between groups. (**A**) Comparison between patients according to the presence of a sequence found as an initiator of recombination (CCCAG); (**B**) Comparison between patients regarding the total presence of recombination motifs. BZLF1: BamHI Z left frame; LMP1: latent membrane protein 1; DRIM: DNA recombination-inducing motif; T: thymine; G: guanine; A: adenine; C: cytosine; ns: not significant (*p* > 0.05); * *p* ≤ 0.05; ** *p* ≤ 0.01.

## Data Availability

All new *BZLF1* sequences obtained in this study were submitted to the GenBank database. The assigned accession numbers are PP056160-PP056229.

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
