# Peer review of "Lytic and Latent Genetic Diversity of the Epstein–Barr Virus Reveals Raji-Related Variants from Southeastern Brazil Associated with Recombination Markers"

_ijms, 2024, doi:10.3390/ijms25095002_

Round 1

Reviewer 1 Report

Comments and Suggestions for Authors

Title: Lytic and latent genetic diversity of the Epstein–Barr virus reveals Raji–related variants from southeastern Brazil associated with recombination markers

General comments: The manuscript by Alves et al. is a significant contribution to the field as it details the genetic diversity of EBV in Southeastern Brazil. The authors have meticulously provided a detailed bioinformatics analysis of the samples, including limitations of their analysis and considerations. Minor comments are mentioned below before the recommendation for acceptance:

  1. Figures: The manuscript effectively describes the results, but all figure quality needs improvement for better readability. Specifically, the letters in the phylogenetic tree, especially the tip labels, should be more legible.
  2. In the materials methods section, MCMC phylogenetic reconstruction was mentioned to be carried out using 100 million steps. What about independent runs/analyses? How many times were the analyses performed? Were the trees in the figures annotated and visualized from a combined independent analysis or representative of a single independent run?
  3. Line 74: remove ' . ' in "...a recent study from our group [15]."

Author Response

Dear reviewer, thank you very much for taking time to carefully review our manuscript. We have answered your questions by topic as follows:

Comments and Suggestions for Authors

Title: Lytic and latent genetic diversity of the Epstein–Barr virus reveals Raji–related variants from southeastern Brazil associated with recombination markers

General comments: The manuscript by Alves et al. is a significant contribution to the field as it details the genetic diversity of EBV in Southeastern Brazil. The authors have meticulously provided a detailed bioinformatics analysis of the samples, including limitations of their analysis and considerations. Minor comments are mentioned below before the recom- mendation for acceptance:

1. Figures: The manuscript effectively describes the results, but all figure quality needs im- provement for better readability. Specifically, the letters in the phylogenetic tree, especially the tip labels, should be more legible.

We apologise for the mistake. All figures have been replaced with better quality.

2. In the materials methods section, MCMC phylogenetic reconstruction was mentioned to be carried out using 100 million steps. What about independent runs/analyses? How many times were the analyses performed? Were the trees in the figures annotated and visualized from a combined independent analysis or representative of a single independent run?

We thank you for the commentary. We carried out several analyses, increasing the number of steps progressively in order to check whether the topology of the tree was maintained, once maintained, we carried out a single run of 100 million steps. This choice was made by choosing the largest possible steps in which the analysis becomes reproducible, not making the analysis extremely long, but generating a robust tree.

3. Line 74: remove ' . ' in "...a recent study from our group [15]."

We apologise for the error, and we have corrected it.

Please see the attachment to the new corrected version of manuscript.

Yours sincerely,

Dr. Paula Alves

Reviewer 2 Report

Comments and Suggestions for Authors

1. The quality of Figures 1, 3, 4, and 5 is poor, making them difficult to view clearly.

2. The introduction and discussion sections are overly verbose and lack clarity.

3. The authors' information should be presented in English.

Comments: 

  1.  Why did the authors select BZLF1 to study EBV genetic diversity? Are there other genes that are highly related to EBV genetic diversity?
  2. It is unclear that why LMP1 was selected to do multigene phylogenetic analysis (MLA) . What about other oncogenes?
  3. Is EBV lytic activated in the 70 individuals?

Author Response

Dear reviewer, thank you very much for taking time to review our manuscript. We have answered your questions by topic as follows:

Comments and Suggestions for Authors

1. The quality of Figures 1, 3, 4, and 5 is poor, making them difficult to view clearly.

We apologise for the mistake. All figures have been replaced with better quality.

2. The introduction and discussion sections are overly verbose and lack clarity.

We thank you for the suggestion. The introduction and discussion were modified to reduce the text and to make it more fluid and clearer.

3. The authors' information should be presented in English.

We apologise for this. The authors’ information has been modified.

Comments:

1. Why did the authors select BZLF1 to study EBV genetic diversity? Are there other genes that are highly related to EBV genetic diversity?

2. It is unclear that why LMP1 was selected to do multigene phylogenetic analysis (MLA) . What about other oncogenes?

Response to the comment 1 and 2: We thank you for the commentary. In fact, other genes, mainly latent ones are more highly related to studies of the EBV genetic diversity due to the greater impact of positive selection in these proteins, such as the EBNAs gene family, that characterize EBV genomes in types in 1 and 2 (Palser et al. 2015, DOI: 10.1128/JVI.03614- 14). However, lytic targets also have considerable variation, and has an impact on the genetic diversity of EBV (Palser et al. 2015, DOI: 10.1128/JVI.03614-14). As shown recently, the V3 variant of the BZLF1 promoter zone (Zp) when present in an EBV type 1 strain was associated with an increased risk of developing disease to confer a functional increase in viral lytic reactivation (Bristol et al. 2018, DOI: 10.1371/journal.ppat.1007179). In this present work, the BZLF1 gene was chosen due our previous study (Alves et al. 2022, DOI: 10.3390/v14081762), where different haplotypic relationship of the EBV type, Zp variants and LMP1 variants was observed. Due the absence of EBV genomes from these samples, we chose to study, through multigene analysis, two highly variable genes within their classes, latent and lytic, since these classes have different evolutionary pressures within the EBV genome. Further, the LMP1 gene was chosen not only because it is one of the most variable and studied EBV latent genes, but because of the practicality of already having the sequences of the same samples.

3. Is EBV lytic activated in the 70 individuals?

Our study design and objectives did not seek to evaluate lytic activity at the time of sample collection. But we understand that the presence of EBV in the blood of patients with post-transplant viral reactivation, and EBV in the saliva of asymptomatic carriers, are sites that suggest activated viral replication. In the case of lymphoma biopsies, even if we performed gene expression, lytic activation would not be expected, since tumors associated with EBV have a profile of a set of active latent genes and not lytic ones (Farrell et al. 2019, DOI: 10.1146/annurev-pathmechdis-012418-013023). Therefore, in the case of neoplasms, the contribution of genetic variation of lytic targets would be more associated with the pathogenesis of the disease (Bristol et al. 2018, DOI: 10.1371/journal.ppat.1007179).

Please see the attachment to the new corrected version of manuscript.

Yours sincerely,

Dr. Paula Alves

Reviewer 3 Report

Comments and Suggestions for Authors

Review of Manuscript “Lytic and latent genetic diversity of the Epstein–Barr virus reveals Raji–related variants from southeastern Brazil associated with recombination markers“ by Paula D. Alves et al..

The authors report on the phylogenetic analysis of the lytic gene BZLF1 in a sampling of 70 EBV-positive cases from southeastern Brazil. This analysis indicated two main clades denoted BZ-A and BZ-B with BZ-A being the more diverse clade associated with the main polymorphisms under investigation. Interestingly, in contrast to the respective single-gene phylogenies, a multigene phylogenetic analysis between BZLF1 and the latency-associated gene LMP1 revealed specific clusters indicating haplotypic segregation. In addition, the authors evaluated the number of DNA recombination-inducing motifs (DRIM) within the different MLA defined clusters, with haplotype BZ-A+Raji containing the highest numbers of both TGGAG and CCCAG motifs. The CCCAG was also the only motif associated with the clinical outcome.  

Due to the nature of the analysis the manuscript is in large parts mostly descriptive. However, the different kind of analyses are well described and clearly presented. At least in the pdf file available for review, however, the resolution of some of the figures was too low (see also major point below). The authors should also streamline the discussion section to significantly shorten the manuscript. This section contains many repetitions of the results section, again in a mostly descriptive manner. The focus of this section should be more on presenting possible implication of the results. 

Major points:

1) The resolution of the figures in the pdf-file is too low, especially evident for figures 1, 3, 4 and 5.

Minor points: 

1) Lines 105 to 107: Abbreviations for group of patients should be explained.

2) Line 370: Typo, change ‘strain’ to ‘strains’.

Comments on the Quality of English Language

Minor editing of English language required

Author Response

Dear reviewer, thank you very much for taking time to review our manuscript. We have answered your questions by topic as follows:

Comments and Suggestions for Authors

Review of Manuscript “Lytic and latent genetic diversity of the Epstein–Barr virus reveals Raji–related variants from southeastern Brazil associated with recombination markers“ by Paula D. Alves et al.

The authors report on the phylogenetic analysis of the lytic gene BZLF1 in a sampling of 70 EBV-positive cases from southeastern Brazil. This analysis indicated two main clades de- noted BZ-A and BZ-B with BZ-A being the more diverse clade associated with the main polymorphisms under investigation. Interestingly, in contrast to the respective single-gene phylogenies, a multigene phylogenetic analysis between BZLF1 and the latency-associated gene LMP1 revealed specific clusters indicating haplotypic segregation. In addition, the authors evaluated the number of DNA recombination-inducing motifs (DRIM) within the different MLA defined clusters, with haplotype BZ-A+Raji containing the highest numbers of both TGGAG and CCCAG motifs. The CCCAG was also the only motif associated with the clinical outcome.

Due to the nature of the analysis the manuscript is in large parts mostly descriptive. However, the different kind of analyses are well described and clearly presented. At least in the pdf file available for review, however, the resolution of some of the figures was too low (see also major point below). The authors should also streamline the discussion section to significantly shorten the manuscript. This section contains many repetitions of the results section, again in a mostly descriptive manner. The focus of this section should be more on presenting possible implication of the results.

We thank you for the commentary. We apologise for the mistake of the figures resolution; all figures have been replaced with better quality. Also, the discussion topic was modified to reduce the text and to make it more fluid and clearer.

Major points:

1) The resolution of the figures in the pdf-file is too low, especially evident for figures 1, 3, 4 and 5.

We apologise for the mistake. All figures have been replaced with better quality.

Minor points:

1) Lines 105 to 107: Abbreviations for group of patients should be explained.

We apologise for the error and have corrected it.

2) Line 370: Typo, change ‘strain’ to ‘strains’.

We apologise for the error and have corrected it.

Please see the attachment to the new corrected version of manuscript.

Yours sincerely,

Dr. Paula Alves

Round 2

Reviewer 2 Report

Comments and Suggestions for Authors

The authors addressed all my comments. I hope the authors will be able to study relationships between other EBV proteins and the genetic diversity of EBV.